# Intraspecies Variation Offers Potential to Improve White Rot Fungi for Increasing Degradability of Lignocellulose for Ruminants

**DOI:** 10.3390/jof10120858

**Published:** 2024-12-11

**Authors:** Anton S. M. Sonnenberg, Nazri Nayan, John W. Cone, Arend F. van Peer

**Affiliations:** 1Plant Breeding, Wageningen University & Research, 6708 PB Wageningen, The Netherlands; nazri.nayan@live.com (N.N.); arend.vanpeer@wur.nl (A.F.v.P.); 2Department of Animal Science, Faculty of Agriculture, Universiti Putra Malaysia, Serdang 43400 UPM, Malaysia; 3Animal Nutrition Group, Wageningen University & Research, 6708 WD Wageningen, The Netherlands; john.cone@wur.nl

**Keywords:** white rot fungi, interspecies variation, rumen degradability, breeding, lignocellulose

## Abstract

The aim of fungal treatment of organic matter for ruminants is the improvement of its degradability. So far, such treatment appears to be time-consuming and improvement has been modest. In previous work, we observed within three white rot species that there is modest (*Ceriporiopsis subvermispora*) or low (*Lentinula edodes* and *Pleurotus eryngii*) variation in fiber degradation in wheat straw during seven weeks of incubation. By extending and re-examining the data from all three species, we see that strains of *C. subvermispora* show the largest variation and improvement in the degradability of treated wheat straw. In addition, *C. subvermispora* also generated the highest absolute amount of degradable organic matter, a parameter not calculated before, but is very relevant for the economic feasibility of fungal treatment. In estimating fungal growth, we found no good correlation between an increase in ergosterol and a decrease in plant biomass, indicating a variation within fungal species of the ergosterol/fungal biomass ratio and/or a variation in carbon use efficiency, which has also not been analyzed before. This work contributes to the knowledge of how fungi degrade lignocellulose and further specifies what can be targeted for breeding to make fungal pretreatment economically feasible for upgrading organic waste streams into ruminal feed.

## 1. Introduction

The enormous amount of biobased carbon on earth is mainly formed by plants [1], and the majority of its mass consists of plant cell walls (PCWs). The presence of the polysaccharides cellulose and hemicellulose makes these PCWs an immense source of raw materials and energy. Next to the polysaccharides, PCWs consist of the heterogeneous aromatic polymer lignin [2]. The density of PCWs and especially the presence of lignin makes enzymatic degradation difficult. White rot fungi (WRF) are known for their ability to degrade complex PCWs [3] and use heme peroxidases to generate low-molecular-weight oxidants that can enter the dense cell walls and break bonds within lignin [4], obtaining access to the energy-rich polysaccharides in this way. Recently, it was demonstrated that WRF are also able to modify and utilize lignin degradation products as a carbon source [5], underlining the prominent role these fungi play in mineralization of carbon, especially in soils [6,7].

Their unique ability to degrade lignocellulose has been identified as a way to pretreat organic matter to obtain access to cellulose for different purposes such as the production of fermentable sugars and bioenergy [8,9] and for the generation of ruminant feed [10]. The pretreatment of organic matter with WRF is, however, still not an economically feasible method to upgrade organic waste. This is due to the fact that the pretreatment with fungi takes a considerable time (weeks) and the enrichment in access to cellulose or the increase in degradability for ruminants is modest [9,11,12,13]. Up to now, most research has focused on comparing fungal species for their selectivity and extent of lignin degradation [8,9,14]. There is, however, evidence that there is also a wide variation between strains within WRF species on the extent and selectivity of lignocellulose degradation [15,16]. Such variation within a species would offer opportunities to improve these fungal traits by breeding.

In a previous study [16], we performed a broad screen, estimating the variability in rumen degradability of wheat straw following treatments with thirty strains of three different WRF species; 12 *Ceriporiopsis subvermispora* (*Gelatoporia subvermispora*), eight *Lentinula edodes* and 10 *Pleurotus eryngii* strains. Changes in the degradability of wheat straw were determined for all 30 treatments using the in vitro gas production (IVGP) method [17], while fungal growth was estimated by measuring the increase in ergosterol. In our previous study, ergosterol content during the seven weeks of incubation on wheat straw was plotted per species, while, in this study, we plotted this per strain. The weak correlation between an increase in ergosterol content and the degradation of organic matter (for *C. subvermispora*) and the absence of this expected correlation (for *L. edodes* and *P. eryngii strains)* raises important questions on aspects of ergosterol concentrations as markers for fungal biomass, the relation of fungal biomass and lignocellulose degradation and thus suitability as selection criteria for fungal treatment. While our previous study conducted organic matter degradability and fiber analyses for three strains per species, we completed fiber analyses for all 30 strains in this study. This allows us not only to assess the change in rumen degradability of organic matter by fungal incubation for each strain but also to determine the net improvements of degradability for all strains. This net improvement is of major importance for economic feasibility. This complementary and more detailed analysis showed that while IVGP was increased by strains of both *L. edodes* and *C. subvermispora*, only strains of the species *C. subvermispora* generated substantial amounts of degradable organic matter. Particularly, the variations within this species offer the opportunity for improvement by breeding.

## 2. Materials and Methods

The samples used here consisted of freeze-dried powder originating from our previous research [16], and details on their origins are given in Appendix A.

The data on ergosterol increase in time during incubation on wheat straw for all strains were used from the previous research [16] but now plotted per strain. We performed fiber analysis (ANKOM Technology, Macedon, New York, USA) for all 30 strains and expressed the degradable organic matter as a percentage of the dry matter and as the absolute amount of digestible organic matter for all 30 strains.

All samples were carried out in triplicate, and extreme outliers (2× SD) were removed. The substrate preparation, fungal inoculations, sampling, ergosterol measurements, fiber analysis (ANKOM) and assessment of IVGP were performed as described in our previous publication [16].

After the IVGP, the remaining content of the bottles was filtered and dried in an oven at 103 °C to determine the dry matter (DM) and in a furnace at 550 °C to determine the ash content. The organic matter (OM) degradability was calculated as a percentage of the digested OM remaining after IVGP assay. The absolute amount of DM, OM and digestible OM was calculated by processing the container content after the fungal incubation as described above. Fiber analyses were performed as described previously [16] but, in addition, each fraction (neutral detergent fiber (NDF), acid detergent fiber (ADF) and acid detergent lignin (ADL)) was corrected for the ash content using the method described above. Hemicellulose was calculated as the difference between NDF and ADF and cellulose as the difference between ADF and ADL.

The T = 0 samples did not contain the spawn inoculum. The correction for DM and OM of the T = 0 samples was made by measuring these parameters for pure spawn and adding the relative amount (corresponding to an inoculum amount of 10% of the dry weight of wheat straw) to the T = 0 samples. The corrections for ergosterol content were performed by using ergosterol measurement performed on pure spawn by Kuijk [18].

Statistical analyses were performed using SPSS 28.

## 3. Results

### 3.1. Ergosterol Concentration in Time

Ergosterol is a membrane component of fungi and is often used to estimate fungal biomass in complex substrates and natural environments [19,20,21]. We previously analyzed [16] the ergosterol content per species during the incubation period and the ergosterol content per strain at week seven to estimate the change in fungal biomass. Here, we plotted the ergosterol data of the same samples in more detail, i.e., per strain within each species during the incubation period (Figure 1A). In addition, a reanalysis revealed that in the previous paper, the ergosterol data for *P. eryngii* for weeks 5 and 7 were switched, and they are now plotted in the correct order. For most strains of *C. subvermispora*, the ergosterol increases during the first five weeks and levels off between weeks five and seven. For *L. edodes*, all strains have a lag phase of three weeks and a continuous increase in ergosterol content from week three to week seven. Most strains of *P. eryngii* show a considerably slower increase in ergosterol content, with the exception of strains Pe4 and Pe5, which clearly show a much stronger increase from weeks one to five and level off between weeks five and seven.

The difference between strains is especially prominent during week seven for the species *C. subvermispora* and *P. eryngii* (Figure 1B). The four highest-ranking strains of *C. subvermispora* generated significantly more ergosterol than the six lowest-ranking strains, and only strain Cs3 differs significantly from all other strains. The two highest-ranking strains of the species *P. eryngii* formed significantly more ergosterol after seven weeks than all other strains of this species. Within the species *L. edodes*, the variation in ergosterol is small and is not significant for most strains (Appendix A; Figure 1B).

### 3.2. Fiber Analyses of Fungal Treated Wheat Straw

After seven weeks of incubation, all strains of the three species affected the fiber composition of wheat straw (Figure 2). For the ADL fraction, there is a significant strain effect within the species *C. subvermispora* and *P. eryngii* (Appendix A). The *C. subvermispora* strains were the most effective in degrading ADL, whereas the *P. eryngii* strains were the least effective. The *C. subvermispora* strains show the largest variation in reducing the absolute amount of the ADL (varying from 33 to 52%). The six best-performing strains are represented by this species and they removed 46–52% of the absolute amount of ADL. The reduction in the absolute amount of ADL in wheat straw by the *P. eryngii* strains varies between 15 and 43% (the outlier strain Pe3 with an unlikely increase in ADL has been removed). The *L. edodes* strains form a middle group, with a reduction in the ADL fraction ranging from 36 to 43%, but without significant variation. As for ADL, the reduction in the absolute amount of hemicellulose (NDF-ADF) variation is seen with a significant strain effect in the species *C. subvermispora* and *P. eryngii*. The *C. subvermispora* and *L. edodes* strains are the species evidently removing the most hemicellulose, with the variation in the reduction between strains ranging from (significant) 48–59% to (non-significant) 42–57%, respectively. The *P. eryngii* strains show the least reduction in hemicellulose (20–35%). The absolute reduction in cellulose (ADF-ADL) is small and is 10% or less for most strains. The strain effect is significant for the species *C. subvermispora* and *L. edodes* (Appendix A).

### 3.3. Effect of Fungal Treatment on Relative and Absolute Amount of Degradable Biomass

Most strains of *C. subvermispora* cause a decrease in the in vitro degradability of OM in wheat straw after one week of incubation, after which the degradability increases between weeks one and five and levels off for most strains between weeks five and seven (Figure 3A). For *L. edodes* strains, there is no change in the degradability of OM for the first three weeks, after which a constant but moderate increase is seen. This corresponds with the ergosterol content, which also shows a lag phase in the first three weeks. For *P. eryngii* strains, a decrease is seen for the first three weeks and there is a continuous increase from week three to week seven. The decrease in degradability in the first week (*C. subvermispora*) or in the first three weeks (*P. eryngii*) is likely due to the consumption of easily accessible carbohydrates and proteins by these fungi. The variation in degradable organic matter between strains is most prominent for the species *C. subvermispora*, especially after five to seven weeks of incubation. Next to the relative degradability, a user will be interested in the absolute amount of digestible OM generated after a fungal treatment since the total amount of digestible lignocellulose will have an important effect on economic feasibility. We compared, therefore, the change in the relative degradability of organic matter in wheat straw after seven weeks of incubation and the change in the absolute amount of degradable organic matter, expressed as a percentage change relative to the untreated material. All strains of the species *C. subvermispora* and *L. edodes* increase the relative degradability of organic matter in wheat straw, as do most strains of *P. eryngii* (Figure 3B). The increase by *C. subvermispora* strains is considerable, varying from 12.6 to 37.1%, and it is modest for *L. edodes*, varying from 4.7 to 10.6%. The increase for the *P. eryngii* strains is low, varying from −8.1 to 3.8%. The absolute (or net) increase in degradable organic matter, however, is only substantial by treatment with the *C. subvermispora* strains. This increase is low for the *L. edodes* strains and even negative for the *P. eryngii* strains. It is also clear that there is not a perfect correlation between the increase in relative and absolute degradability (Figure 3B). This is most prominent for strains of the species *C. subvermispora,* Especially in the group of C10, Cs4, Cs5 and Cs2, where there is no significant difference between the relative change in degradability, the change in absolute degradability differs significantly between these strains (Figure 3C; Appendix A). For the species *L. edodes,* the change in both types of degradability after seven weeks of incubation was hardly present or not significant between strains. For the species *P. eryngii*, the variation between both types of degradability is also low and most strains do not differ significantly from each other. The change in IVGP during the incubation time obviously has a similar course to the relative and absolute digestible OM, since the former has been used to derive the latter two parameters (Appendix A).

Since fiber composition likely will have an influence on the degradability of the fungal treated wheat straw, we correlated the change in fiber composition (derived from ANKOM data) with the absolute amount of in vitro degradability. Since the strains consume easily degradable components and likely no fibers in the first week, we estimated the correlation between the change in the absolute amount of fibers and the absolute amount of degradable OM from T = 3 up to T = 7 for each species (Table 1). As lignin is considered the main obstacle for microbial and enzymatic degradation of the polysaccharides in PCWs [22], it is not surprising that all species show a significant negative correlation between the absolute amount of ADL and the absolute amount of digestible OM (Table 1A). Also, a significant negative correlation between the degradability of digestible OM and hemicellulose and a significant positive correlation with the absolute amount of cellulose for the species *C. subvermispora* and *L. edodes* were found (Table 1B,C). For all data, the R^2^ value is rather low, indicating a considerable variation in data, i.e., a variation in performance between strains within species.

## 4. Discussion

Although the use of white rot fungi has the potential to increase the degradability of organic matter for ruminants, the process is time-consuming and the actual improvement is rather modest, making this type of pretreatment not economically feasible. In addition to optimizing the technical aspects of such treatments, the improvement of the fungal strains by breeding can be an important way to enhance efficiency. A prerequisite for breeding is the presence of biological variation within species for those characteristics that are relevant for the improvement of the degradability of organic matter for ruminants, i.e., the time needed for pretreatment of organic matter, the selectivity in degrading fiber fractions, and the increase in degradability while retaining as much organic material as possible. The above analyses show that these biological variations are low or not significant between strains within the species *L. edodes* and *P. eryngii* but they are substantial and significant for most strains of the species *C. subvermispora*. The low variation within the *L. edodes* species might be due to the low genetic variation. We do not know the exact origin of the *L. edodes* strains but most are likely cultivars or derived from cultivars. The genetic variation within cultivars is much smaller than for wild strains as is shown, for example, by the diversity of the mating type in commercial shiitake strains, which is 5.5 times lower than in wild isolates [23]. The low variation between the *P. eryngii* strains is likely due to the fact that this species is not very effective at selectively degrading lignocellulosic material, and all strains have a low performance.

The speed and extent of colonization of the substrate by fungi is a relevant phenotype because this will likely influence the time needed for pretreatment and the amount of organic matter degraded. Quick colonization of the substrate is also important to prevent infections by other microorganisms, especially relevant since industrial-scale treatments will not use sterilization. The biological variation in the time needed to colonize the substrate is thus a target for breeding. Ergosterol content is the most used indicator for estimating fungal biomass in complex substrates. However, the ergosterol content can differ considerably between fungal species [19,24]; thus, our ergosterol data cannot be used to compare fungal biomass between the three species used here. The comparison of fungal biomass between strains within one species can also be problematic. Ergosterol content is considered one of the best estimates for fungal biomass but it does not discriminate between living and dead hyphae and the correlation between ergosterol and fungal biomass might not always be linear [19,25,26]. In addition, the ergosterol content of hyphae can vary on different substrates and culture conditions [24,27]. Here, all strains were cultured on the same substrates and under the same environmental conditions, and the change in ergosterol content likely reflects the change in fungal biomass for individual strains in time. For the *C. subvermispora* strains, there is a significant negative correlation between the ergosterol content and the dry matter content of the containers after seven weeks of incubation (Table 2). Fungal biomass is generated at the cost of dry matter. This might indicate a fairly constant ratio in ergosterol content and fungal biomass between strains within this species. A negative correlation between the ergosterol content and the decrease in dry matter degradation, however, is not seen for *L. edodes* and *P. eryngii*. There is a positive correlation between the ergosterol content and the dry matter decrease after seven weeks for *L. edodes* strain and no correlation at all for the *P. eryngii* strains. For all species, the *R^2^* (the goodness of fit of the multivariate regression) is 0.5 or lower, indicating a large variation in the data (Table 2). Visual inspection of the substrate after seven weeks of growth shows that there is some correlation between the whiteness of the substrate seen with the naked eye and ergosterol content for *C. subvermispora* strains, but this correlation is less clear for the other two species (Appendix A). There are two possible explanations for this lack of correlation for the *L. edodes* and *P. eryngii* strains. Firstly, the ergosterol content per unit of mycelial dry weight could vary between strains within a species. Hardly any data has been published comparing the ergosterol content between strains within one fungal species. The change in ergosterol in time might thus be used to estimate mycelial biomass formation for each strain but cannot be used to compare fungal biomass between strains within the same species if the ergosterol/fungal biomass ratio varies within species. As a result, the correlation between the increase in ergosterol and the consumption of the substrate DM does not exhibit the expected negative correlation. A second explanation might be a variation in the carbon use efficiency (CUE) between strains within one species. CUE is defined as the C generated as new microbial biomass divided by the total C taken up from the substrate, with the difference primarily attributed to respiration. The CUE varies considerably between fungal species [28] and is influenced by abiotic factors [29], but nothing is known about the variation in CUE between strains within one fungal species. For future experiments, it will be useful to assess the ergosterol/fungal biomass ratio for each strain in time. These data can also be used to measure variation in the CUE within fungal species. Such data can be generated in experiments where the fungal biomass can be separated from the growth medium as in liquid cultures or solid substrates covered with a semipermeable membrane. Whether such data can be translated directly for growth in complex substrates is, however, uncertain since both the ergosterol content and the CUE depend on the substrate used [6]. Other methods that allow the assessment of fungal components in complex substates might be useful, such as quantitative polymerase chain reaction (qPCR) [30] or the use of transcriptomic markers relevant for fungal growth and respiration [28].

Currently, the ergosterol formation over time does not seem to be a suitable marker for selecting the best species or strains to improve lignocellulose degradability. Moreover, for some strains of *C. subvermispora*, the absolute amount of degradable organic matter is maximal after five weeks of incubations, whereas for other strains, this is seven weeks (Appendix A). It is thus necessary to measure the absolute degradable organic matter generated by each strain during the whole pretreatment period.

The selective degradation of fibers in organic matter by WRF will influence the accessibility of polysaccharides for enzymatic degradation. Recently, direct evidence was presented for a covalent (α-ether) linkage between lignin and hemicellulose [31]. As lignin also plays a key role in the enzymatic degradation of hemicellulose, it is expected that there is a correlation between the decrease in ADL and the decrease in hemicellulose during fungal treatment. This correlation is indeed seen for the species *C. subvermispora* and *L. edodes* (Table 1D). The fact that the *C. subvermispora* strains generate the largest amount of absolute digestible OM after seven weeks is not due to the highest concentration of polysaccharides in the treated straw (Table 3). This might indicate that the *C. subvermispora* strains, in particular, increase the enzymatic accessibility of polysaccharides. In our ANKOM analyses, we measured the insoluble fiber fraction and cannot discriminate between the decrease in the amount of fibers due to fungal consumption and solubilization by fungal enzymes. The actual amount of polysaccharides might thus be higher. Kijpornyongpan and colleagues [32] compared the presence of CAZymes (carbohydrate-active enzymes) in secretomes of ten white rot fungi, which included *C. subvermispora* and *P. eryngii*. They collected data from fungal species for which at least three corresponding data sets were available, with each data set containing at least 50 detected proteins. *P. eryngii* had considerably more CAZymes for all PCWs than *C. subvermispora*, 103 versus 73, respectively. In our present data and previous research [16,33], *C. subvermispora* clearly performs better than *P. eryngii* in degrading lignocellulose. A prediction of the degradation of PCW components from complete and well-annotated genome sequences or even secretomes thus seems to be challenging. This underlines the importance of phenotyping to estimate the extent and selectivity of lignocellulose degradation by fungal strains.

The clear phenotypic variation seen between the *C. subvermispora* strains in increasing the degradability of wheat straw, as well as the different fiber components and ergosterol patterns, indicates potential for improvement by the breeding of fungal strains. In addition, the time for the treatment might be reduced if the time required for complete colonization could be shortened. *C. subvermispora* produces chlamydospores [34], possibly enabling the mass inoculation of organic matter by these spores and reducing the time required for colonization.

## 5. Conclusions

This research shows that within the species *C. subvermispora*, a variation exist for phenotypes that are relevant to improving the accessibility of energy-rich polysaccharides in lignocellulose. The targets for breeding this species are the time needed to completely colonize the substrate and the improvement of degradability while retaining as much organic matter as possible. Thus, breeding has the potential to make this pretreatment economically feasible for upgrading organic waste streams into ruminal feed or feedstock for biotechnological purposes.

## Figures and Tables

**Figure 1 jof-10-00858-f001:**
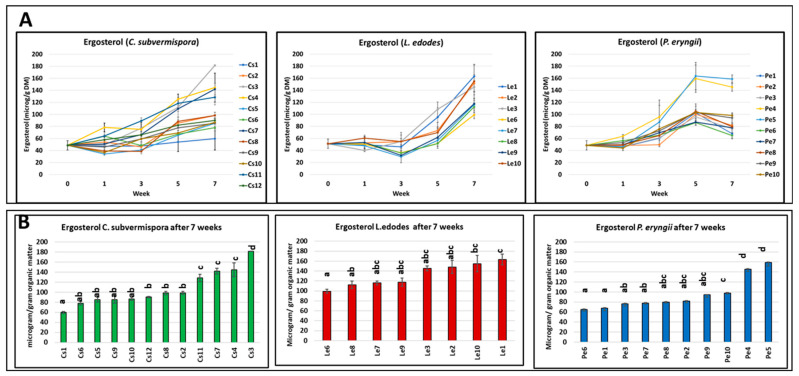
(**A**) Ergosterol concentrations during the incubation for seven weeks of wheat straw with different strains of three white rot fungal species. (**B**) The ergosterol concentration after seven weeks of incubation. Error bars: ± standard deviation. Bars in the graphs with different letters within a column are significantly (*p* < 0.05) different.

**Figure 2 jof-10-00858-f002:**
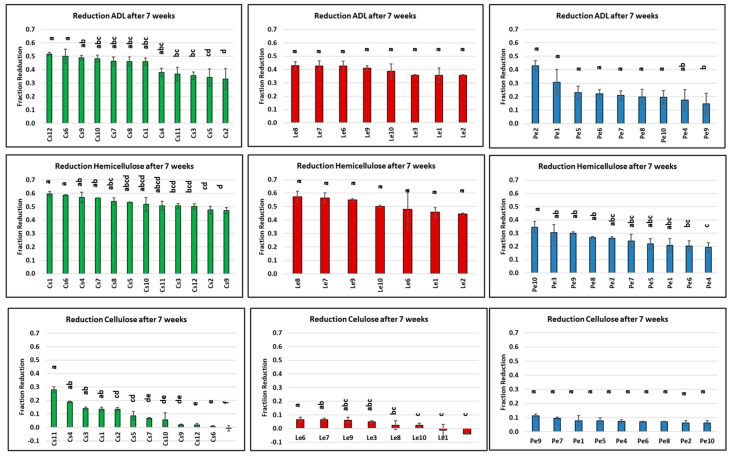
The reduction in fiber (fraction of untreated material) after seven weeks of incubation of wheat straw. Strains of each species are in different colors (Cs: green; Le: red, Pe: blue). ADL: Acid detergent lignin; hemicellulose: NDF-ADF; Cellulose: ADF-ADL. Le3 for reduction hemicellulose was left out because just one measurement was included. Bars with different letters are significantly (*p* < 0.05) different.

**Figure 3 jof-10-00858-f003:**
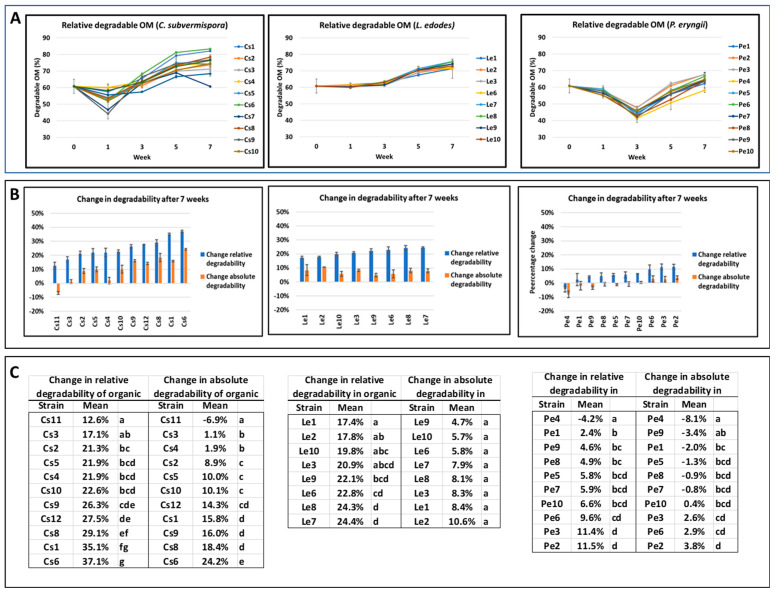
Change in degradability of wheat straw during or after incubation with strains of three different fungal species. (**A**) Change in relative degradability expressed as percentage of degradable organic matter in wheat straw during seven weeks of incubation. (**B**) Change in relative (blue bars) or absolute (orange bars) degradability of wheat straw after seven weeks of incubation expressed as fraction of the untreated material. (**C**) Statistics on panel (**B**). Values with different letters after each third column per species are significantly (*p* < 0.05) different. Cs: *C. subvermispora* (Cs7 left out because of only one measurement being available); Le: *L. edodes*; Pe: *P. eryngii*.

**Table 1 jof-10-00858-t001:** Two-tailed Pearson’s correlation between the absolute amount of different fibers and the absolute amount of degradability of organic matter (DOM) during weeks 3 to 7 of fungal incubation. A: Correlation between absolute amount of DOM and the absolute amount of ADL; B: correlation between the absolute amount of DOM and the absolute amount of cellulose (ADF-ADL); C: correlation between the absolute amount of DOM and the absolute amount of hemicellulose (NDF-ADF); D: correlation between the absolute amount of ADL and the absolute amount of hemicellulose. R^2^ represents the goodness of fit of the multivariate regression.

**Pearson’s correlations between the absolute amount of different fibers and absolute amount of degradability of organic matter DOM) during weeks 3 to 7 of incubation**							
*A* Species	Abs_DOM (g/container) by Abs ADL (g/container)		R^2^	*C* Species	Abs_DOM (g/container) by Abs Hemicellulose (g/container)		R^2^
*C. subvermispora*	Pearson’s Correlation	−0.497 **	0.247	*C. subvermispora*	Pearson’s Correlation	−0.383 **	0.147
	Sig. (2-tailed)	<0.001			Sig. (2-tailed)	<0.001	
	N	94			N	99	
*L. edodes*	Pearson’s Correlation	−0.543 **	0.295	*L. edodes*	Pearson’s Correlation	−0.613 **	0.376
	Sig. (2-tailed)	<0.001			Sig. (2-tailed)	<0.001	
	N	67			N	64	
*P. eryngii*	Pearson’s Correlation	−0.462 **	0.213	*P. eryngii*	Pearson’s Correlation	−0.785 **	0.613
	Sig. (2-tailed)	<0.001			Sig. (2-tailed)	<0.001	
	N	79			N	87	
*B* Species	Abs_DOM (g/container) by Abs Cellulose (g/container)		R^2^	*D* Species	Abs_ADL (g/container) by Abs Hemicellulose (g/container)		R^2^
*C. subvermispora*	Pearson’s Correlation	0.666 **	0.443	*C. subvermispora*	Pearson’s Correlation	0.667 **	0.444
	Sig. (2-tailed)	<0.001			Sig. (2-tailed)	<0.001	
	N	94			N	101	
*L. edodes*	Pearson’s Correlation	0.123	0.015	*L. edodes*	Pearson’s Correlation	0.863 **	0.744
	Sig. (2-tailed)	0.323			Sig. (2-tailed)	<0.001	
	N	67			N	67	
*P. eryngii*	Pearson’s Correlation	0.015	0.0002	*P. eryngii*	Pearson’s Correlation	0.319 **	0.102
	Sig. (2-tailed)	0.899			Sig. (2-tailed)	<0.001	
	N	79			N	82	

** Correlation is significant at the 0.01 level (2-tailed).

**Table 2 jof-10-00858-t002:** Two-tailed Pearson’s correlation between the ergosterol content and the absolute amount of organic matter after 7 weeks of fungal incubation. R^2^ represents the goodness of fit of the multivariate regression.

**Pearson’s correlation between ergosterol concentration and dry matter (DM) content after 7 weeks of fungal incubation**			
Species	Ergosterol (µg/g DM) by Absolute DM (g/container)		
*C. subvermispora*	Pearson’s Correlation	−0.601 **	R^2^
	Sig. (2-tailed)	<0.001	0.361
	N	36	
*L. edodes*	Pearson’s Correlation	0.750 **	0.563
	Sig. (2-tailed)	<0.001	
	N	22	
*P. eryngii*	Pearson’s Correlation	−0.172	0.03
	Sig. (2-tailed)	0.363	
	N	30	

** Correlation is significant at the 0.01 level (2-tailed).

**Table 3 jof-10-00858-t003:** Absolute amount of different fiber fractions (g/container) after seven weeks of fungal incubation of wheat straw. The total polysaccharides (polysacch.) are the sum of hemicellulose and cellulose.

**Absolute amount of different fiber fraction (g/container) after 7 weeks of incubation**								
species	ADL	SD	hemicellulose	SD	cellulose	SD	total polysacch.	SD
*C. subvermispora*	5.97	0.79	12.03	1.16	37.56	3.61	49.62	3.70
*L. edodes*	6.31	0.48	12.82	1.77	40.14	1.68	53.01	2.84
*P. eryngii*	8.00	0.92	19.24	1.45	38.29	0.90	57.53	1.71

## Data Availability

The data presented in this study are openly available in [4TU Repository] at [https://doi.org/10.4121/13ef571d-8d9e-4476-ab58-be3c3b707ad9], accessed on 8 November 2023.

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
