# Peer review of "Intraspecies Variation Offers Potential to Improve White Rot Fungi for Increasing Degradability of Lignocellulose for Ruminants"

_jof, 2024, doi:10.3390/jof10120858_

Round 1

Reviewer 1 Report (Previous Reviewer 1)

1. While the ability of White Rot Fungi (WRF) to degrade lignocellulose and their potential in pre-treating organic matter for use in ruminant feeds was mentioned in the introduction, the connection between the structure of lignocellulose and the degradation mechanism of WRF could have been elaborated on more thoroughly. For instance, references to cellulose, hemicellulose, and lignin in lignocellulose could be expanded by explaining how their chemical structural features influence the degradation process by WRF, rather than merely stating that their presence makes enzymatic degradation challenging.

2. When discussing the relationship between ergosterol content and dry matter degradation in different fungal strains, two possible explanations have been proposed for the lack of correlation in L. edodes and P. eryngii strains. However, the interrelationships or prioritization of these possibilities could be explored further. The authors should clarify whether there is evidence suggesting that changes in ergosterol/fungal biomass ratios are more likely to cause the lack of correlation in these species, or if variations in carbon utilization efficiency (CUE) play a more critical role.

3. Regarding the difference in the number of CAZymes between C. subvermispora and P. eryngii and their actual ability to degrade lignocellulose, a more detailed discussion is needed. It would be helpful to explore whether other studies have similarly observed that fungi with a higher number of CAZymes perform poorly in degrading lignocellulose and to explain the reasons for these discrepancies.

4. Please ensure that the references adhere to the journal’s formatting requirements, as several entries either lack page numbers or contain incorrect details.

5. The language lacks coherence, which impedes smooth comprehension. It is strongly recommended that the authors have the article reviewed by a native speaker to enhance its clarity and logical structure.

1. While the ability of White Rot Fungi (WRF) to degrade lignocellulose and their potential in pre-treating organic matter for use in ruminant feeds was mentioned in the introduction, the connection between the structure of lignocellulose and the degradation mechanism of WRF could have been elaborated on more thoroughly. For instance, references to cellulose, hemicellulose, and lignin in lignocellulose could be expanded by explaining how their chemical structural features influence the degradation process by WRF, rather than merely stating that their presence makes enzymatic degradation challenging.

2. When discussing the relationship between ergosterol content and dry matter degradation in different fungal strains, two possible explanations have been proposed for the lack of correlation in L. edodes and P. eryngii strains. However, the interrelationships or prioritization of these possibilities could be explored further. The authors should clarify whether there is evidence suggesting that changes in ergosterol/fungal biomass ratios are more likely to cause the lack of correlation in these species, or if variations in carbon utilization efficiency (CUE) play a more critical role.

3. Regarding the difference in the number of CAZymes between C. subvermispora and P. eryngii and their actual ability to degrade lignocellulose, a more detailed discussion is needed. It would be helpful to explore whether other studies have similarly observed that fungi with a higher number of CAZymes perform poorly in degrading lignocellulose and to explain the reasons for these discrepancies.

4. Please ensure that the references adhere to the journal’s formatting requirements, as several entries either lack page numbers or contain incorrect details.

5. The language lacks coherence, which impedes smooth comprehension. It is strongly recommended that the authors have the article reviewed by a native speaker to enhance its clarity and logical structure.

Author Response

Reviewer 2. While the ability of White Rot Fungi (WRF) to degrade lignocellulose and their potential in pre-treating organic matter for use in ruminant feeds was mentioned in the introduction, the connection between the structure of lignocellulose and the degradation mechanism of WRF could have been elaborated on more thoroughly. For instance, references to cellulose, hemicellulose, and lignin in lignocellulose could be expanded by explaining how their chemical structural features influence the degradation process by WRF, rather than merely stating that their presence makes enzymatic degradation challenging.

Reply: Elaborating on the degradation mechanism by white rot fungi on lignocellulose is beyond the scope of our manuscript. There is indeed a lot known on this mechanism but also a lot unknown. We refer to this in our discussion where we point out that although Pleurotus eryngii has a higher number of lignolytic and auxiliary genes/enzymes, this species is the lesser performer in generating digestible organic matter. 

Reviewer 2. When discussing the relationship between ergosterol content and dry matter degradation in different fungal strains, two possible explanations have been proposed for the lack of correlation in L. edodes and P. eryngii strains. However, the interrelationships or prioritization of these possibilities could be explored further. The authors should clarify whether there is evidence suggesting that changes in ergosterol/fungal biomass ratios are more likely to cause the lack of correlation in these species, or if variations in carbon utilization efficiency (CUE) play a more critical role.

Reply. We have suggested indeed two possibilities for the lack of correlation between ergosterol content and organic matter degradation. These are possible mechanisms (not prove one) and it is thus impossible to see what mechanism might have the upper hand. That would just be speculation and would not benefit the discussion paragraph.

Reviewer 2. Regarding the difference in the number of CAZymes between C. subvermispora and P. eryngii and their actual ability to degrade lignocellulose, a more detailed discussion is needed. It would be helpful to explore whether other studies have similarly observed that fungi with a higher number of CAZymes perform poorly in degrading lignocellulose and to explain the reasons for these discrepancies.

Reply. Again, this discrepancy between the number of CAZymes and the ability to degrade lignocellulose is an observation that point to the fact that on the base of the number of enzymes one cannot predict the effect of a species the increasing effect of degradability. To elaborate on this, we would have to study a number of genomes of other basidiomycete and see if there are more examples that the one we have shown. That is also beyond the scope of our manuscript and we think it will not contribute more to the point we have already made.

Reviewer 2. Please ensure that the references adhere to the journal’s formatting requirements, as several entries either lack page numbers or contain incorrect details.

Reply. Some corrections were made. For some references only link is available and no page number.

Reviewer 2. The language lacks coherence, which impedes smooth comprehension. It is strongly recommended that the authors have the article reviewed by a native speaker to enhance its clarity and logical structure.

Reply. We have asked a native speaker to checked the manuscript. This person was fine with the text. We have published a considerable number of papers in English and this is the first time I have had such a comment.

Reviewer 2 Report (Previous Reviewer 2)

The paper by Sonnenberg and collegues presents the comparison of differents strains from three fungi. It is interesting to reanalyze the data obtained form a previous study to bring new information. However, I think the presentation of these results should be improved. The main conclusion is that differences between strains in the capacity to modify organic matters for ruminant feeding exist almost only for the C. subvermispora fungus, as very few significant variations were observed for the two other fungi (L. edodes and P eryngii). The authors explain that the lack of differences for the strains of these fungi could be explained by the low genetic variation (beggining of discussion). Moreover, as these fungi have no variation between strains, the presentation at the end of the paper return to the comparison of the species (lines 168-189), and not the strains between them. For these reasons, I strongly suggest the authors to present only the data about C. submermispora to make a clear case. It could allow to clarify the data presented throughout the paper. As it is currently, the excess of data dilute the main message and make it harder to understand. Eventually, the data of L. edodes and P eryngii could be presented in supporting data. 

Figure 1. If the authors choose to keep all data, the scales of the graphs (B) should be homogeneous for comparison. Also, there are discrepancies between the supplementary data (statisticals) and the corresponding figure 1B (letters regarding the differents groups).

Figure 2. The comparison of each parameters (ADL, hemicellulose and cellulose) should be presented side by side to compare the fungi species (if they keep the three species). Similar comments to fig 1B/ S2 can be done for the statistics (S3). The presentation of S3 should be improved as the title is on the last sheet.

Overall, the presentation of the tables is not acceptable. They look like screen copies of Excel table. Morevoer, the data corresponding to table 1 and table 2 should be presented as graphs. 

The discussion presents interesting aspects, but if the authors want to have a title related to strains/species improvement, they should clearly describe which strains among those they evaluate could be advantageous for ruminants feeding, and/or which are the most promising for breeding. I suggest the authors to choose a title more realted to thier finding and keep the "potential" of their findings for application in the discussion.

see main comments

Author Response

Reviewer 1: The paper by Sonnenberg and collegues presents the comparison of differents strains from three fungi. It is interesting to reanalyze the data obtained form a previous study to bring new information. However, I think the presentation of these results should be improved. The main conclusion is that differences between strains in the capacity to modify organic matters for ruminant feeding exist almost only for the C. subvermispora fungus, as very few significant variations were observed for the two other fungi (L. edodes and P eryngii). The authors explain that the lack of differences for the strains of these fungi could be explained by the low genetic variation (beggining of discussion). Moreover, as these fungi have no variation between strains, the presentation at the end of the paper return to the comparison of the species (lines 168-189), and not the strains between them. For these reasons, I strongly suggest the authors to present only the data about C. submermispora to make a clear case. It could allow to clarify the data presented throughout the paper. As it is currently, the excess of data dilute the main message and make it harder to understand. Eventually, the data of L. edodes and P eryngii could be presented in supporting data. 

Reply: A number of remarks are helpful and we will make these corrections.

We like to emphasize once more, and have clearly indicated throughout the manuscript, that previous data were not only reanalyzed, but also extended allowing new conclusions.
We have carefully addressed all the remarks from reviewer #1 in our previous submission, but we see that the reviewer now has made new objections/directions. One is the suggestion of the reviewer to present not the result for all three species but only for the species with the most significant relevant variations in parameters. We state indeed that for the species L. edodes and P. eryngii the variation of some (but not all!) parameters are low. For the species L. edodes we hypothesize that this might be due to a low genetic variation. That is, however, not certain, just a possible explanation which is of importance to consider when one is planning to screen and compare strains within a species. The low observed variation of the P. eryngii strains is certainly not due to low genetic variation, but has (likely) to do with the overall poor performance of this species on the tested substrates regarding the investigated parameters. In our opinion it is thus relevant to show results for all species and that this study might indicate (but not proof) that the strains of C. subvermispora not only perform best, but also show the largest phenotypic variation, a prerequisite for breeding.

Reviewer 1: The discussion presents interesting aspects, but if the authors want to have a title related to strains/species improvement, they should clearly describe which strains among those they evaluate could be advantageous for ruminants feeding, and/or which are the most promising for breeding. I suggest the authors to choose a title more realted to thier finding and keep the "potential" of their findings for application in the discussion.

Reply: Our group is one of the expertise centers in breeding  mushrooms. For breeding, a variation in phenotype in a collection of strains for breeding is an important prerequisite. The existence of this prerequisite is what we demonstrate in our paper. But also the genetics of the life cycle, such as recombination in meiosis, and dominance or expression of phenotypes in subsequent generations are used important in breeding. Identifying the best strains for the use in breeding is usually not done by just screening phenotypes but first by estimating the breeding value of selected phenotypic traits by crossing all possible selected strains and measuring their performance. That will make clear what strains will reveal the largest phenotypic segregation in offspring and potentials for mapping genomic regions underlying the traits of interest. This is, however, beyond the scope of this manuscript. Stating what strains have to be used for breeding should thus not be made based on just their performance in a first screening.
The title of the manuscript states, in our opinion, correctly that intraspecific variation offers potentials for breeding.

Reviewer 1: Figure 1. If the authors choose to keep all data, the scales of the graphs (B) should be homogeneous for comparison. Also, there are discrepancies between the supplementary data (statisticals) and the corresponding figure 1B (letters regarding the differents groups).

Reply: The corrections are made for figure 1.

Reviewer 1: Figure 2. The comparison of each parameters (ADL, hemicellulose and cellulose) should be presented side by side to compare the fungi species (if they keep the three species).

Reply: We have grouped fiber types in columns, one way to make the comparison. But we now changed this by grouping fiber type by row, and hope that is what the reviewer meant to be done.

Reviewer 1: Similar comments to fig 1B/ S2 can be done for the statistics (S3). The presentation of S3 should be improved as the title is on the last sheet.

The title has been added to page 1.

 Reviewer 1: Overall, the presentation of the tables is not acceptable. They look like screen copies of Excel table. Morevoer, the data corresponding to table 1 and table 2 should be presented as graphs. 

Reply: The final layout of table are usually done by the final editing of the manuscript. We have adapted the tables and hopefully they are now acceptable.

Round 2

Reviewer 2 Report (Previous Reviewer 2)

The authors answered my comments.

 The authors answered my comments.

This manuscript is a resubmission of an earlier submission. The following is a list of the peer review reports and author responses from that submission.

Round 1

Reviewer 1 Report

 This study examined the Intraspecies variation in white rot fungi for increasing degradability of lignocellulose. The results are somewhat beneficial for understanding the variation of selectivity in lignin degradation within species, which unveils a previously overlooked phenomenon yet might exist in the natural environment. But The article is not well written. There are a lot of errors.

1. This study examined the Intraspecies variation in white rot fungi for increasing degradability of lignocellulose. The results are  somewhat beneficial for understanding the variation of selectivity in lignin degradation within species, which unveils a previously overlooked phenomenon yet might  exist in the natural environment. But The article is not well written. There are a lot of errors.

2. The authors poorly describe the innovation and significance of this research work. What can be targets for breeding to make fungal pretreatment economic feasible?

3. The English writing need to be carefully polished. E.g., show a much stronger increase

4. What is the relationship between substrate pH and degradability of lignocellulos?

5. How was the Ergosterol quantified? It is unclear from the methods section.

Reviewer 2 Report

In the paper untitled “Intraspecies variation offers potentials to improve white rot fungi for increasing degradability of lignocellulose for ruminants.” Sonnenberg and coworkers compare numerous strains a three different fungal species to grow and degrade lignocellulose. This work is the direct continuation of the paper by Nayan et al (ref 16: Nayan, N.; Sonnenberg, A. S.; Hendriks, W. H.; Cone, J. W., Screening of white-rot fungi for bioprocessing of wheat straw into ruminant feed. Journal of applied microbiology 2018, 125 (2), 468-479.).

I consider this manuscript is not suitable for publication unless strong modifications and clarification are made.

I am concerned about potential self-plagiarism. Indeed, although the authors clearly state that the sample were those from Nayan et al (ref 16). It is not clear whether the data presented in figs 1, 2 and 3 are new or reused. This should be clearly state, and the interest of reusing the data should be explained. See for example my next comment about the pH measurements.

Introduction: lines 48-51, introduce the concept of “strains”. Although they were presented in Nayan et al. 2018, the origine of the strains and their differences should be presented here, at least in supplementary data.

Introduction, line 52: the reference 16 by Nayan et al. refers to previous work form the authors. I suggest to the authors to modify the sentence to show that this new work is the following of a previous work by the team. More generally, I suggest replacing “Nayan et al” by “we” throughout the manuscript.

 Paragraph 3.2. “Effect of strains on substrate pH”. Does this part necessary to be included in the manuscript? The results are not presented considering the different strains, by the 3 species. Are there differences between the strains of one species? If not, this part could be removed or shortened and described in supplementary data. Are these results presenting something different than those already presented in ref 16? This seems to be very similar to me. The figure 2 is very unclear – very likely because the legend is lacking. What are the dots? What are the “times”? Actually, most of legends are incomplete (Figs. 2, 3, 4, 6).

Paragraph 3.4 (that should be 3.3). “Fiber analyses of fungal treated wheat straw”. The authors constantly refer to Supplementary data S1, but the test and the significance of the results should appear in the figure 3. The strains should be compared between the species, and not altogether. We don’t know which strains are significantly more or less efficient than the others. Once again, are these results different than those presented in the table 4 of the ref 16? Moreover, the sentence “Only four C. subvermispora strains consume more cellulose, i.e. 13-28%.” (line 151) is very unclear – “more the cellulose” than what?

-          Paragraph (which has no number) “Effect fungal treatment on relative degradability and absolute digestible biomass”. I think that Figure 5 should be a panel c of the figure 4, as it just corresponds to the point obtained at week 7. But, once again the strains should be compared by species and the statistics should appear – which strains are significantly more or less efficient than the others?

-          The tables 1 and 2 present statistical data comparing the species. I don’t understand how the calculation was made. For each species, which strains were considered? Apparently, all and an average was made, explaining the low R² values. But why doing that? The goal of the paper is to evaluate if there are differences between the strains of a single species. I really suggest removing all the comparison of species throughout the manuscript, and focus only on the comparison of strains. Considering the figure 4, we can see that there are differences between the strains of the species of C. subvermispora but apparently not for the strains of the other species.

-          Discussion: line 317, “It is remarkable that the four C. subvermispora strains with the lowest performance in generating digestible OM represent those that have the highest ergosterol content and the highest DM degradation after seven weeks of incubation.” We don’t see that at all in the figure 6, and this would not be consistent with the apparent correlation.

-          Discussion: the main finding of the paper is finally written only at the very end of the discussion: “This research has shown that within the species C. subvermispora a variation exist for phenotypes relevant for improving the accessibility of the energy-rich polysaccharides in lignocellulose.” This should be described before, and the main part of the discussion might be about the fact that there are no differences between the strains of the other species. This main result should be in the abstract.

-          Supplementary data S1 is very unclear. The authors should present all their supplementary files according to the scientific standard, in a text/word file, with required title and legends, containing sufficient explanations. Overall, the presentation and formatting are not good. The figures are of bad quality and legends are lacking or incomplete. The correct formatting was very likely not checked before the submission.

-          

Abstract: please, replace “OM” by “organic matter”

-          Please, check the English, some sentence could be improved – ex: replace “economic feasible methods” by “economically feasible methods” (in the abstract, introduction).

-          Line 111, this sentence is unclear: “For most strains of C. subvermispora the ergosterol increases during the first five weeks and levels off for most strains between week five and seven.” Please, rephrase.

-          Line 284: there is no “Table 2D”.

-          Line 291: define “CAZymes”. Not all readers know the meaning of this acronym.